# Seroprevalence and Risk Factors of Lyme Borreliosis in The Netherlands: A Population-Based Cross-Sectional Study

**DOI:** 10.3390/microorganisms11041081

**Published:** 2023-04-20

**Authors:** B. J. A. Hoeve-Bakker, Oda E. van den Berg, H. S. Doppenberg, Fiona R. M. van der Klis, Cees C. van den Wijngaard, Jan A. J. W. Kluytmans, Steven F. T. Thijsen, Karen Kerkhof

**Affiliations:** 1Centre for Infectious Disease Control, National Institute for Public Health and the Environment (RIVM), 3721 MA Bilthoven, The Netherlands; 2Department of Medical Microbiology and Immunology, Diakonessenhuis Hospital, 3582 KE Utrecht, The Netherlands; 3Department of Medical Microbiology, University Medical Center Utrecht, Utrecht University, 3584 CX Utrecht, The Netherlands

**Keywords:** *Borrelia*, epidemiology, C6 Lyme ELISA, immunoblot, multivariable analysis, serology, standard two-tier testing, surveillance

## Abstract

Lyme borreliosis (LB) is not notifiable in many European countries, and accurate data on the incidence are often lacking. This study aimed to determine the seroprevalence of *Borrelia burgdorferi* sensu lato (s.l.)-specific antibodies in the general population of The Netherlands, and to determine risk factors associated with seropositivity. Sera and questionnaires were obtained from participants (n = 5592, aged 0–88 years) enrolled in a nationwide serosurveillance study. The sera were tested for *B. burgdorferi* s.l.-specific IgM and IgG antibodies using ELISA and immunoblot. Seroprevalence was estimated controlling for the survey design. Risk factors for seropositivity were analyzed using a generalized linear mixed-effect model. In 2016/2017, the seroprevalence in The Netherlands was 4.4% (95% CI 3.5–5.2). Estimates were higher in men (5.7% [95% CI 4.4–7.2]) than in women (3.1% [95% CI 2.0–4.0]), and increased with age from 2.6% (95% CI 1.4–4.4) in children to 7.7% (95% CI 5.9–7.9) in 60- to 88-year-olds. The seroprevalence for *B. burgdorferi* s.l. in the general population in The Netherlands was comparable to rates reported in European countries. The main risk factors for seropositivity were increasing age, being male and the tick bite frequency. The dynamics of LB infection are complex and involve variables from various disciplines. This could be further elucidated using infectious disease modelling.

## 1. Introduction

Spirochetes of the *Borrelia burgdorferi* sensu lato (s.l.) complex are the causative agent of Lyme borreliosis (LB), which is the most prevalent tick-borne disease in the northern hemisphere. In Europe, these spirochetes are transmitted by ticks of the *Ixodes ricinus* complex [1]. Of these, at least five *Borrelia* genospecies are human pathogens: *B. afzelii*, *B. garinii*, *B. burgdorferi* sensu stricto, *B. bavariensis*, and *B. spielmanii*. After transmission during tick attachment, infection starts with localized manifestations, such as erythema migrans (EM) or borrelial lymphocytoma. If left untreated, LB may progress towards more debilitating manifestations including Lyme neuroborreliosis (LNB), Lyme arthritis, Lyme carditis and acrodermatitis chronica atrophicans. In The Netherlands, the total disease burden of LB was estimated at 10.6 disability-adjusted life years per 100,000 population [2].

Understanding the epidemiology of LB is essential to make rational health planning decisions to reduce the disease burden. However, estimation and comparison of the incidence of LB are complicated, as surveillance is not standardized in the European Union (EU). In an earlier study, various scenarios for surveillance of LB were explored, of which the surveillance of LNB seemed the most feasible way to standardize LB surveillance in the EU, supplemented with surveillance of EM [3]. As a result of this study, LNB was added to the EU list of communicable diseases under surveillance [4]; however, in many European countries, including The Netherlands, LNB surveillance has not been implemented thus far. Therefore, nationwide data on the incidence of LNB and other LB manifestations are often scarce and/or incomplete regarding demographic data such as age, gender, and place of residence [5].

In contrast to the incidence, seroprevalence may be considered a cumulative registry of past, present and newly acquired infections. Hence, serosurveillance could also contribute to a better understanding of the epidemiology of LB. For accurate representation of nation-wide seroprevalence rates and identification of local hotspots, population-based cohorts should be used [6]. As this approach is very expensive and time consuming, studies applying such strategies are rare.

In The Netherlands, three national biobanks for population-based cross-sectional seroprevalence studies (PIENTER) have been established since 1995 [7,8,9]. Although primarily aiming to evaluate the seroprevalence of vaccine-preventable diseases included in the National Immunization Programme, these biobanks have also been used for serosurveillance of various infectious diseases, such as Q-fever [10] and cytomegalovirus [11]. In this study, the third biobank (PIENTER-3; established in 2016–2017) has been used to study the seroprevalence of *B. burgdorferi* s.l.-specific antibodies in the general population in The Netherlands and to identify associated risk factors.

## 2. Materials and Methods

### 2.1. Study Population

The third national biobank for population-based cross-sectional seroprevalence studies (PIENTER-3) was established in 2016/2017 and described in detail by Verberk et al. [9]. This biobank includes a national sample obtained from 40 randomly selected municipalities distributed in five geographic regions in The Netherlands. An age-stratified sample was drawn from each municipality’s population register. An oversampling of non-Western migrants living in The Netherlands was added to this sample, as this group was underrepresented in the initial sample. Additionally, a subpopulation of individuals living in areas with low vaccination coverage (LVC) was included. A venous blood sample was obtained from 6151 participants from the national sample (n = 4977) and the LVC (n = 1174) (Figure 1).

### 2.2. Laboratory Analysis

The national sample and LVC sample of the PIENTER-3 biobank contained 6151 sera samples (Figure 1). Of these, 546 sera samples were not analyzed due to insufficient volume. The remaining 5605 sera samples were tested for *B. burgdorferi* s.l.-specific immunoglobulin (Ig) G and IgM antibodies using the standard two-tier testing strategy that is also recommended for clinical diagnosis of LB. The sera were screened using the C6 Lyme ELISA (Immunetics, Boston, MA, USA), which uses a synthetic 26-mer peptide derived from the sixth invariable region (IR6) of the VlsE protein [12]. The assay was automatically processed on a Freedom Evo 200 pipetting robot (Tecan Group Ltd., Männedorf, Switzerland), according to the manufacturer’s instructions. Negative screening results were considered seronegative. Positive and equivocal screening results were confirmed with immunoblot (IB) analysis using the recomLine Borrelia IgG and IgM immunoblots (Mikrogen, Neuried, Germany), following the manufacturer’s instructions. Negative IgM and IgG IB results and equivocal IgM IB results were considered seronegative. Equivocal IgG results, positive IgM and IgG IB results were considered seropositive.

### 2.3. Data Analysis

Data management and analyses were conducted in R version 4.2.1 [13].

The questionnaires were designed for the evaluation of the National Immunization Programme [9] and not all questions were relevant for this study. Therefore, a subset of the questions was selected by each of the authors, and used for data analyses.

Upon merging of laboratory data and questionnaire data, 13 participants were excluded due to missing questionnaire data (Figure 1). The remaining participants that were included in further analysis belonged to the national sample (n = 4567) and LVC (n = 1025).

Sociodemographic characteristics and variables possibly related to LB were described for participants of the national sample and LVC. Seroprevalence estimates (with 95% Confidence Intervals [CI]) for *B. burgdorferi* s.l.-specific antibodies were calculated from the national sample using R’s survey package [14], considering the survey design (i.e., controlling for geographic region [Nielsen] and municipality) and weighted by sex, age, ethnic background and degree of urbanization to match the distribution of the general Dutch population.

Risk factor analysis was performed to determine behavioral and intrinsic factors associated with *B. burgdorferi* s.l. seropositivity. For this analysis, data from the national sample and LVC samples were used, as the sample group was not associated with seropositivity. Risk factor analysis was performed in various steps, starting with a univariable analysis using logistic regression modelling with R’s lme4 package [15]. A generalized linear mixed-effect model, including a random intercept to account for the survey design (i.e., potential clustering by municipality and geographic region [Nielsen]) was optimized using Akaike’s Information Criterion (AIC). Univariable analysis was carried out using the optimized model fit on selected explanatory variables from the questionnaires using binary serology results (positive or negative) as outcome variables, and a priori adjustment for age and gender. Variables that reached a significance level of *p*-value < 0.10 in the univariable analyses were included in the multivariable analysis. For multivariable analysis the model was optimized through stepwise backward deletion based on Akaike’s Information Criterion (AIC). Adjusted odds ratios (aOR) with 95% CIs were reported.

## 3. Results

### 3.1. Study Population

Only participants with sufficient serum and questionnaire data available from the PIENTER-3 biobank were included, resulting in 5592 participants, of whom 4567 (81.7%) belonged to the national sample and 1025 (18.3%) belonged to the LVC. In the national sample, more females than males were included (55.0% vs. 45.0%); most participants were born in The Netherlands (78.9%) and did not report a tick bite in the last 5 years (73.7%) (Table 1). Participants were equally distributed with regards to region of residence and educational degree; however, participants from areas with a very low degree of urbanization were slightly underrepresented.

### 3.2. Laboratory Analysis

The sera from the PIENTER-3 biobank were screened for *B. burgdorferi* s.l.-specific IgG and IgM antibodies. In 4830 (86.4%) of the 5592 sera, no *B. burgdorferi* s.l.-specific antibodies were detected using the C6 Lyme ELISA. The remaining 762 (13.6%) reactive sera were subjected to IB analysis. For 528 (69.3%) of 762 sera, a negative IB result was reported for both IgG and IgM. Of the 234 (30.7%) reactive IB results, 24 (3.1%) were positive IB for IgM only, 48 (6.3%) were equivocal for IgG only, 139 (18.2%) were positive for IgG only, two (0.3%) sera were equivocal for IgM and positive for IgG, and 21 (2.8%) sera were positive for both isotypes (Appendix A).

### 3.3. Seroprevalence Estimates

*B. burgdorferi* s.l.-specific antibodies were detected in 187 of the 4567 sera from the national sample, resulting in a weighted seroprevalence of 4.4% (95% CI 3.5–5.2) (Table 1). The seroprevalence estimate was higher in males than in females (5.7% vs. 3.1%). The seroprevalence increased with age, ranging from 2.6% in the 0–19 years age group to 7.7% in the 60–88 years age group (Figure 2). The seropositivity per sampled municipality ranged from 0.7% to 10.2% (Figure 3). Regional differences in seroprevalence ranged from 3.7% in the southwestern provinces to 5.2% in the northeastern provinces of The Netherlands and was the highest in areas with a very low degree of urbanization (8.0%). Seroprevalence was higher among participants with Dutch (4.7%) or other Western (4.8%) nationalities than among participants from non-western descent (1.2% to 3.2%). The seroprevalence increased with the number of tick bites reported in the last 5 years and ranged from 3.4% among participants that reported no tick bites to 23.5% among participants that reported ten or more tick bites.

### 3.4. Risk Factor Analysis

Univariable analyses were performed on all selected variables with a priori adjustment for age and gender (Appendix A). In the multivariable analysis, variables that were associated with seropositivity included gender, age, country of birth, educational level, number of tick bites (last 5 years), number of antibiotic therapies (last 3 months), eating pork meat and eating raw vegetables (Table 2).

Multivariable analysis revealed that men were more at risk of seropositivity than women (aOR 1.78; 95% CI 1.35–2.35), and odds on being seropositive increased with age up to 3.97 (95% CI 2.55–6.20) for 60- to 88-year-olds (Table 2). Significantly higher odds were observed for those that reported one or more tick bites in the last five years, up to an aOR of 10.01 (95% CI 4.41–22.73) for participants reporting ten or more tick bites in the five years prior to inclusion, in comparison to participants that reported no tick bites in the last five years. Additionally, participants that were born in a foreign country had lower odds of being seropositive than participants that were born in The Netherlands. Participants with a high (higher vocational secondary education and university education) level of education had slightly (but not significantly) higher odds of being seropositive. Lower odds were observed for participants that used antibiotics within three months prior to inclusion compared to those that did not, although CIs included one, indicating non-significance. Interestingly, eating pork three or more times per week appeared to be protective for seropositivity (aOR 0.32; CI 0.13–0.80) and participants that ate raw vegetables three or more times per week had higher odds of being seropositive (aOR 1.76; CI 1.08–2.86).

## 4. Discussion

This study showed that in 2016/2017, the prevalence of *B. burgdorferi* s.l. antibodies in the general population in The Netherlands was 4.4% (95% CI 3.5–5.2). Estimates were higher in men than in women, and increased with age and tick bite frequency.

In other northwestern European countries, the seroprevalence ranged from 1.1% in Belgium [6] to 24% in parts of Sweden [16]. This broad range can be partly explained by the various assays and testing strategies used on different types of study populations. Additionally, ecological factors may play a role in the observed variation in seroprevalence rates. For instance, local density of competent hosts and vectors as well as the prevalence of *B. burgdorferi* s.l. in ticks [17] influence human exposure and thus seroprevalence. In addition, a study from Finland reported that current seroprevalence rates are much lower than half a century ago, which seems in contrast to the reported increase in ticks, tick bites and LB in Finland and in many other countries in the last two decades [18]. In The Netherlands, although the observed regional differences in seroprevalence were marginal, both seroprevalence and EM incidence [19] were the highest in the northeastern provinces in The Netherlands.

For serosurveillance of LB, the demands of the testing strategy are met in the recommended standard two-tier testing strategy for LB serodiagnosis: a highly sensitive screening assay to reduce false-negative test results, and confirmation of positive test results using a highly specific assay. In clinical practice, laboratory testing for *B. burgdorferi* s.l.-specific serum antibodies is carried out after careful assessment of the pretest probability [20], and 63.0% of the reactive sera in the C6 Lyme ELISA were also positive in the confirmation assay in the Diakonessenhuis hospital during 2016–2017. This is in line with previous results obtained in a hospital setting [21]. Here, the standard two-tier testing strategy was applied on an asymptomatic study population (without assessment of pretest probability), and only one-third of the positive screening results were confirmed using immunoblot. This ratio is in concordance with those found for asymptomatic populations in previous studies [21,22], and demonstrates the need for confirmation of the screening results in a serosurveillance setting. When compared to a hospital setting, twice as many positive screening results are not confirmed using immunoblot, and this clearly demonstrates the added value of two-tier testing in a serosurveillance setting.

The age-seroprevalence curve for the general population in The Netherlands was S-shaped, starting with an increase during childhood, stabilization at 3–4% between 20 to 50 years of age, and again increasing to 9% in 80-year-olds. Therefore, no less than 9% of the general population in The Netherlands becomes infected with *B. burgdorferi* s.l. at least once during their lifetime. This is probably an underestimation since seroreversion, which is also boosted with antibiotic treatment [23], is not included in this estimate. Additionally, a similar S-shaped distribution was observed in the German population [24] and might possibly be explained by different risk behavior in different age groups.

Risk factor analysis revealed increasing age, being male and increasing number of tick bites as the main variables predicting increased seropositivity. To our knowledge, this is the first population-based study that reports a highly significant association with tick-bite frequency; however, age and gender were previously associated with seropositivity [6,16,25]. As clearly demonstrated in a longitudinal study, a 0.8% increase in seroprevalence could be attributed to the aging population in Germany [24]. Future research with the focus on birth cohorts in a longitudinal study design will need to address this.

The observed association between seropositivity and eating pork meat or raw vegetables was unexpected. These findings were not previously reported and any possible causality could not be inferred from the available data. Similarly, having a pet cat was associated with increased seropositivity in German children [25]; however, having a cat or any other pet was not a significant risk for seropositivity in our study. This once again demonstrates that knowledge of the mechanisms influencing LB infection is far from complete.

The strengths of this study include the use of a cross-sectional population-based study population that allows for accurate seroprevalence estimation for the general population in The Netherlands and identification of regional hotspots. Additionally, although IgG is generally the isotype tested for in seroprevalence studies [6,24], immunoblot confirmation using IgM was also included in this study. The inevitably observed IgM-only seropositive results may be explained by (i) early infection at the time of sampling, (ii) abrogated isotype-switching due to spirochetal clearance from effective antibiotic treatment [26,27], (iii) persistent IgM response [28] or (iv) false-positive reaction due to cross-reacting IgM antibodies [23,29].

This study has some limitations. As in all other cross-sectional population-based studies, the study population may have suffered from selection bias, i.e., people with specific characteristics that are associated with LB serostatus, such as age or sex, might unintentionally be overrepresented in the study population. To limit the influence of possible selection bias, the seroprevalence estimates were weighted by sex, age, ethnic background and degree of urbanization to match the distribution of the general Dutch population. Additionally, the applied sampling strategy, i.e., two-stage clustering based on geographic region and municipality, may also have resulted in selection bias, which was corrected for in both the seroprevalence estimates and the regression models.

As demonstrated here and in other studies, the dynamics of LB seroprevalence are complex and involve many variables from various disciplines. For instance, seroconversion and reversion rates obtained from longitudinal datasets [24] and underlying variables such as the use of antibiotics and the type of antigens were used in the assay [23]. Additionally, ecological information on host and tick density, *B. burgdorferi* s.l. prevalence in both, and transmissibility of *B. burgdorferi* genotypes can be illuminating. Modelling all these variables could be a powerful tool to gain insight in disease pathology and epidemiology of LB. Infectious disease modelling in which multiple factors are combined is already more advanced for other infectious diseases, and was particularly successful using longitudinal data [30]. Although initial efforts have been made [31], infectious disease modelling for LB is still in its infancy.

In conclusion, the seroprevalence for *B. burgdorferi* s.l. in the general population in The Netherlands is 4.4%, and the main risk factors for seropositivity are increasing age, being male and the number of tick bites reported. The dynamics of LB infection are complex and involves many variables from various disciplines, and could be further elucidated using infectious disease modelling. 

## Figures and Tables

**Figure 1 microorganisms-11-01081-f001:**
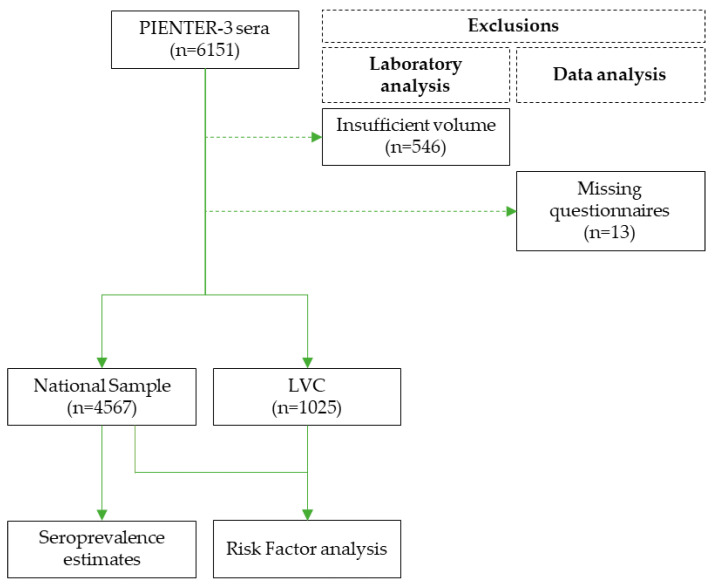
A flow chart of the study setup. The national sample and the Low Vaccination Coverage sample (LVC) of the PIENTER-3 biobank [9] were used to study the seroprevalence and risk factors of Lyme borreliosis in The Netherlands in 2016–2017. The national sample was used for seroprevalence estimates. Both the national sample and the LVC were used for risk factor analysis.

**Figure 2 microorganisms-11-01081-f002:**
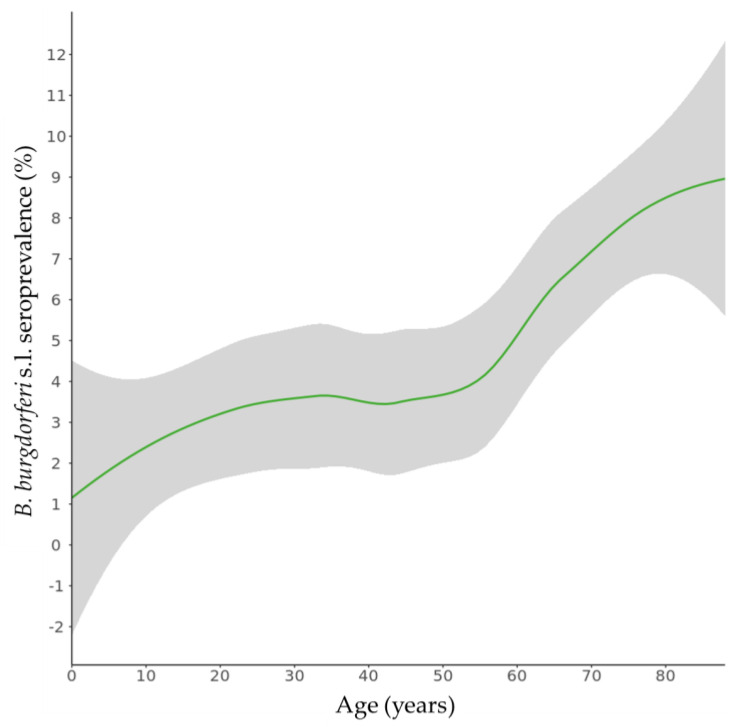
Smooth age-specific seroprevalence of *B. burgdorferi* sensu lato (s.l.) antibodies in the general population of The Netherlands. Grey areas indicate 95% confidence intervals.

**Figure 3 microorganisms-11-01081-f003:**
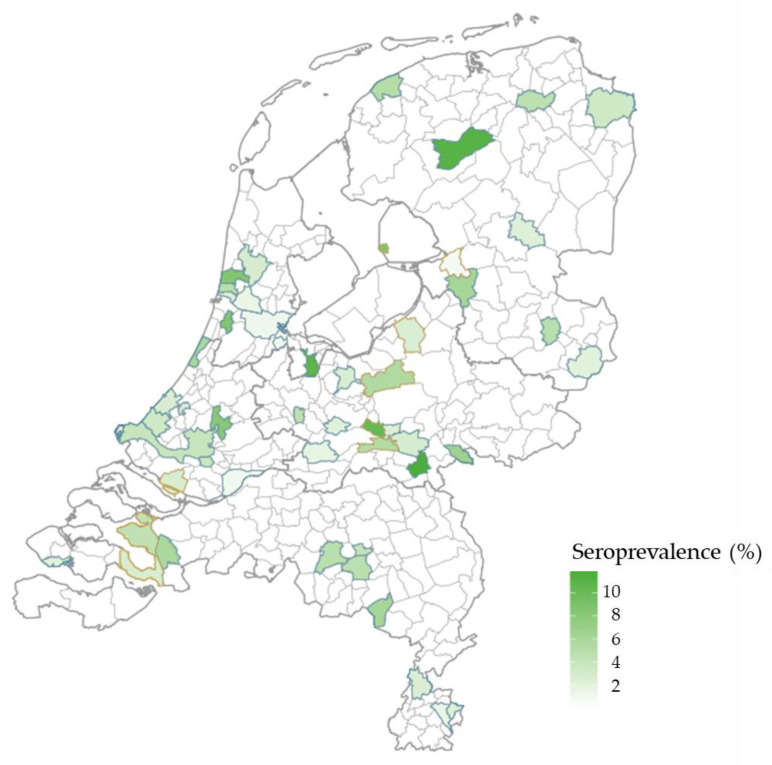
Lyme borreliosis seropositivity in the sampled municipalities in The Netherlands. Blue and orange boundaries indicate municipalities in the national sample and low vaccination coverage sample, respectively.

**Table 1 microorganisms-11-01081-t001:** Sociodemographic characteristics of the participants and weighed seroprevalence of anti-*B. burgdorferi* sensu lato antibodies in the general population of The Netherlands.

	National Sample				Low NIP Vaccination Coverage Sample	
	Total		Weighted Anti-*Borrelia* Antibodies Seroprevalence		Total	
	n	%	%	95% CI	n	%
**Overall**	4567	100	4.4	3.5–5.2	1025	100
**Gender**						
Female	2511	55.0	3.1	2.3–4.0	610	59.5
Male	2056	45.0	5.7	4.4–7.2	415	40.5
**Age (years)**						
0–19	840	18.4	2.6	1.4–4.4	205	20.0
20–39	1392	30.5	3.6	2.2–5.5	359	35.0
40–59	1188	26.0	3.6	2.5–5.1	265	25.9
60–88	1147	25.1	7.7	5.9–9.7	196	19.1
**Region ^1^**						
Northeast	916	20.1	5.2	2.6–9.0	NA	NA
Southeast	1135	24.9	4.2	3.0–5.6	NA	NA
Central	847	18.5	4.2	1.8–8.2	NA	NA
Southwest	870	19.0	3.7	2.1–5.9	NA	NA
Northwest	799	17.5	4.8	2.2–8.9	NA	NA
LVC municipalities	NA	NA	NA	NA	1025	100
**Degree of urbanization (addresses per km^2^)**						
Very high (>2500)	988	21.6	4.3	2.5–6.8	0	0
High (1500–2500)	1480	32.4	4.1	2.6–6.0	0	0
Medium (1000–1500)	884	19.4	4.0	1.9–7.2	128	12.5
Low (500–1000)	829	18.2	4.0	2.4–6.2	560	54.6
Very low (0–500)	386	8.5	8.0	0.0–57.5	337	32.9
**Ethnicity**						
Dutch	3604	78.9	4.7	3.7–5.8	981	95.7
Other Western	317	6.9	4.8	2.6–7.9	29	2.8
Moroccan and Turkish	112	2.5	3.2	0.1–15.1	3	0.3
Surinamese, Aruban, The Netherlands Antillean	223	4.9	1.2	0.3–3.6	4	0.4
Other non-Western	311	6.8	2.5	1.0–5.0	8	0.8
**Level of education ^2^**						
Low	1262	27.6	4.1	2.7–6.0	615	60.0
Middle	1480	32.4	3.1	2.2–4.1	143	14.0
High	1559	34.1	5.7	4.3–7.5	211	20.6
**Number of tick bites in the last 5 years**						
Never	3364	73.7	3.4	2.6–4.3	765	74.6
1–4 times	565	12.4	8.2	5.5–11.8	131	12.8
5–9 times	42	0.9	8.8	2.3–21.7	9	0.9
≥10 times	28	0.6	23.5	9.9–42.7	7	0.7
Unknown	292	6.4	4.6	2.6–7.4	67	6.5

CI, confidence interval; LVC, Low Vaccination Coverage sample; NIP, National Immunization Programme. ^1^ Regions comprised the following provinces; Northeast: Groningen, Friesland, Drenthe and Overijssel, Southeast: Noord-Brabant and Limburg, Central: Gelderland and Utrecht, Southwest: Zeeland and Zuid-Holland, Northwest: Noord-Holland and Flevoland. ^2^ For participants < 15 years, the educational level of the mother was used. Missing: National sample: educational level, 266; net monthly income, 682; no. of tick bites in the last 5 years, 276; low vaccination coverage sample: educational level, 56; net monthly income, 141, number of tick bites in the last 5 years, 46.

**Table 2 microorganisms-11-01081-t002:** Risk determinants for seropositivity.

		n	n_pos_	%	aOR [95% CI]
**Gender**					
Female		3121	99	3.2	Reference
Male		2471	135	5.5	1.78 [1.35–2.35]
**Age (years)**					
0–19		1045	28	2.7	Reference
20–39		1751	51	2.9	1.19 [0.73–1.92]
40–59		1453	51	3.5	1.52 [0.94–2.46]
60–88		1343	104	7.7	3.97 [2.55–6.20]
**Number of tick bites in the last 5 years**					
Never		4129	129	3.1	Reference
1–4 tick bites		696	54	7.8	2.52 [1.79–3.55]
5–9 tick bites		51	5	9.8	2.92 [1.10–7.76]
≥10 tick bites		35	9	25.7	10.01 [4.41–22.73]
Unknown		359	21	5.8	1.84 [1.13–2.99]
**Country of birth**					
The Netherlands		4928	219	4.4	Reference
Turkey and Morocco		69	1	1.4	0.22 [0.03–1.65]
SAN		177	4	2.3	0.41 [0.15–1.17]
Other		418	10	2.4	0.45 [0.23–0.89]
**Level of education ^1^**					
Low		1623	62	3.8	Reference
Middle		1877	61	3.2	0.96 [0.66–1.41]
High		1770	95	5.4	1.40 [0.98–2.01]
**Use of antibiotics in the last 3 months**					
Never		4284	182	4.2	Reference
1 time		422	17	4.0	0.92 [0.54–1.54]
2 times or more		120	1	0.8	0.16 [0.02–1.14]
**Eating pork meat**					
Never		538	25	4.6	Reference
<1 day/month		463	25	5.4	1.07 [0.59–1.94]
1–3 days/month		1220	53	4.3	0.78 [0.47–1.30]
1–3 days/week		2636	103	3.9	0.69 [0.43–1.11]
>3 days/week		302	6	2.0	0.32 [0.13–0.80]
**Eating unwashed vegetables**					
Never		2051	85	4.1	Reference
<1 day/month		635	39	6.1	1.66 [1.10–2.49]
1–3 days/month		946	30	3.2	0.87 [0.56–1.35]
1–3 days/week		1137	41	3.6	1.03 [0.69–1.53]
>3 days/week		474	24	5.1	1.76 [1.08–2.86]

aOR, adjusted Odds Ratio; CI, confidence interval; SAN, Suriname, Aruba, The Netherlands Antilles. ^1^ For participants < 15 years the educational level of the mother was used. Missing: number of tick bites in the last 5 years, 322; educational level, 322; use of antibiotics in the last 3 months, 766; eating pork meat, 433; eating unwashed vegetables, 349.

## Data Availability

All the relevant data are included in the manuscript.

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
