# Peer review of "Seroprevalence and Risk Factors of Lyme Borreliosis in The Netherlands: A Population-Based Cross-Sectional Study"

_microorganisms, 2023, doi:10.3390/microorganisms11041081_

Round 1

Reviewer 1 Report

This interesting study used an available, nationally representative biobank of samples from individuals in the Netherlands, and performed standard 2 tier Lyme testing to estimate national seroprevalence rates. I have just a few queries and comments.

1.     Could the authors describe how samples are selected for inclusion in the biobank to assure this is a random, unbiased sampling?

2.     In describing 2 tier testing, it states equivocal IgM and IgG blots were counted as seropositive.  What were the definitions of positive and equivocal?  How many of the samples were equivocal vs positive? To what extent did inclusion of equivocal samples drive results? What is the justification for this?

3.     Line 127 states the sample was 55% male, 45% female. Table 1 states the reverse

4.     Line 151 states 187 sera were considered seropositive, equal to the number with a positive IgG blot only. Subsequent discussion (line 251) states both IgG and IgM blot+ individuals were included.  Please clarify 

5.     Statistical significance: The text states significant variables included number of tick bites and use of antibiotics in preceding 3 months. Regarding tick bites, the CIs of the 4 groups seem to overlap considerably.  Can the basis of the statistically significant difference be shown more clearly? Regarding antibiotics, the CIs all include 1.  Please clarify regarding statistical significance.

6.     Given that samples were all (presumably) obtained from asymptomatic individuals, false positives become an important concern – emphasizing the relevance of considering ‘equivocal’ western blots as positive.  Please clarify.

7.     The assumption (line 233) that positive serology reflects recent exposure seems suspect – is it really more likely that individuals 60-88 are being exposed to and bitten by ticks than those 40-59 or younger?  Unless there is compelling evidence supporting this assumption, or concerning the rate at which seropositive individuals become seronegative, I would consider this assumption mere speculation.

Reviewer 2 Report

Dear authors,

I appreciate the opportunity to help improve the study, and hope that suggestions are welcome:

- Keywords have terms that are in the title. The ideal would be to replace it with other terms that would help index the study.

- Line 32: The abbreviation of the genus Borrelia and sensu lato need not be in parentheses. These abbreviations are established taxonomic rules.

- Line 36: The abbreviation of sensu stricto need not be in parentheses. These abbreviations are established taxonomic rules.

- Line 47: What is EU?

- Line 79: Figures together with their legends must be understood independently of the main text. So the legend of Figure 1 needs to be improved. Study of what? Study where? Study When? This data needs to be in the legend.

- In Table 1, there are abbreviations, and because it is a table, they must be clarified in the caption or below.

- In general, the text contains many abbreviations. It's hard to read. Abbreviations should be used to facilitate reading, but when they occur in excess, the reader is lost. At various times, I forgot what a certain abbreviation meant and looking for it in the text to find out what it was about became a tedious job.

- One aspect was not clearly raised by the authors, especially in the discussion, about age as a risk factor. In the case of a seroprevalence study, does age reflect on the time of exposure to pathogens, why was this not explored? Despite few studies on borreliosis, on several other pathogens that have advanced age as a risk factor for seroprevalence, we know that the older the individual, the more time there was for exposure. This can be explored in the discussion as well.

In addition to the notes raised, the study is excellent, with precious and important data.

Kind Regards.

Round 2

Reviewer 1 Report

Thank you for your responses

Author Response

The pleasure was all ours. Thank you again for the time and effort spent on our manuscript.